# Application of the STRATCANS Criteria to the MUSIC Prostate Cancer Active Surveillance Cohort: A Step Towards Risk-Stratified Active Surveillance

**DOI:** 10.3390/cancers17183032

**Published:** 2025-09-17

**Authors:** Ana M. Moser, Michael Wang, Ava Zamani, Sabir Meah, Stephanie Daignault-Newton, Corinne Labardee, Nicholas Dybas, Jacob Clapper, Brian R. Lane, Tudor Borza, Alice Semerjian, Vincent J. Gnanapragasam, Kevin B. Ginsburg

**Affiliations:** 1Department of Urology, Wayne State University, Detroit, MI 48202, USA; 2Department of Urology, University of Michigan, Ann Arbor, MI 48109, USA; 3Department of Urology, Capital Urology Associates, Marshall, MI 49068, USA; 4Division of Urology, Corewell Health West, Grand Rapids, MI 49506, USA; 5Department of Surgery, Division of Urology, University of Cambridge, Cambridge CB2 0QQ, UK

**Keywords:** active surveillance, follow-up, prostate cancer, risk-stratified, STRATCANS, urologic oncology

## Abstract

Active surveillance is a common approach for men with low-risk prostate cancer, allowing them to avoid or delay surgery and radiation while closely monitoring their condition. However, there is no clear agreement on how often these men should have check-ups, blood tests, imaging, or biopsies, leading to wide variation in monitoring frequency and intensity between physicians. The STRATified CANcer Surveillance (STRATCANS) model, developed at the University of Cambridge, tailors follow-up by grouping patients into one of three levels of risk based on their clinical features, with each group having a different recommended monitoring schedule. In this study, we applied STRATCANS to a large cohort of men in the Michigan Urological Surgery Improvement Collaborative registry to assess its ability to predict disease progression and the need for treatment in a diverse, real-world setting. Our findings could support more personalized care, reduce unnecessary procedures, and improve patients’ quality of life.

## 1. Introduction

Prostate cancer (PC) is the most common solid tumor diagnosed in men [1,2,3]. The use of active surveillance (AS) as the primary treatment modality for favorable-risk PC in the United States is increasing, with up to 90% utilization in low-risk cases due to favorable long-term oncologic outcomes and lower morbidity than definitive therapies like prostatectomy and radiation [4,5,6,7,8,9]. Although long-term studies have shown AS to be safe, how we should perform AS remains unclear [10,11,12]. As a result, the frequency of lab tests, imaging, biopsies, and follow-up among men on AS varies widely among providers [7,13,14,15,16]. Establishing an appropriate AS approach is crucial to avoid missing the window of cure for more aggressive disease, while also ensuring that lower-risk patients are not overburdened, potentially leading to financial toxicity or discontinuation of AS itself [17,18,19]. The lack of a clear risk-based follow-up schedule makes it difficult for physicians to determine the appropriate monitoring intensity for AS patients [20,21]. To address this, Gnanapragasam and colleagues from the University of Cambridge developed the STRATified CANcer Surveillance (STRATCANS) criteria [22,23]. STRATCANS risk-stratifies patients appropriate for AS into one of three tiers (STRATCANS 1, 2, or 3) with differing follow-up schedules based on prognostic classification factors at diagnosis including clinical stage, PSA, PSA density, and the National Institute of Health and Care Excellence (NICE) Cambridge Prognostic Groups (CPG), which are derived from grade groups (GG). STRATCANS was built based on the risk of progression to ≥CPG3 (unfavorable intermediate-risk PC or higher), diagnosed by upgrade on biopsy to ≥GG3 or upstage to ≥T3 on imaging. In a prospective trial, treatment for progression to ≥CPG3 was 4.7% in STRATCANS 1 (least intense follow-up), compared to 12.9% and 27.4% in STRATCANS 2 and 3 (most intense follow-up), respectively, over a 5-year period [17]. The use of STRATCANS reduced clinic follow-up visits, repeat MRI scans, biopsy events, and associated resource costs [18].

STRATCANS was developed in UK and Spanish cohorts and further validated in a UK cohort [17,18]. These men had a standardized diagnostic and management pathway, including high utilization of pre-biopsy MRI. To date, STRATCANS has not been tested in a large, diverse cohort to assess its real-world generalizability. If STRATCANS is associated with patient outcomes using data from the Michigan Urological Surgery Improvement Collaborative (MUSIC) Prostate registry, this would support a risk-adapted, personalized, follow-up strategy instead of a one-size-fits-all approach, potentially lowering healthcare costs, reducing patient burden, and improving patient adherence to follow-up schedules.

We applied the STRATCANS criteria to the MUSIC Prostate AS cohort to examine the association between STRATCANS tiers (STRATCANS 1, 2, and 3) with the risk of biopsy upgrading and time to definitive treatment.

## 2. Materials and Methods

### 2.1. Study Population and Design

This is a retrospective review of PC patients on AS in the MUSIC registry. MUSIC is a quality improvement consortium involving over 260 urologists across 46 diverse practices in the state of Michigan [24]. This analysis includes men diagnosed with GG1 and GG2 PC from 2016 to 2022 who were managed with AS. Consistent with previously established criteria in MUSIC, AS was defined as the affirmative selection of AS as the primary management strategy, with no treatment received within 6 months of diagnosis. Patients were excluded if they did not meet this criterion, if they were missing variables needed to determine STRATCANS tier (see below), had metastasis on imaging within 6 months of diagnosis, lacked data to calculate PSA density, or had follow-up < 6 months. Each practice obtained exemption or approval from their local Institutional Review Board (IRB) for participation in the MUSIC collaborative. This study was deemed exempt from review by the Wayne State University IRB.

### 2.2. STRATCANS

STRATCANS risk-stratifies patients into three tiers: STRATCANS 1, 2, and 3—with STRATCANS 1 having the least and STRATCANS 3 the highest risk of biopsy progression. STRATCANS prognostic classification is based on factors at the time of diagnosis.

STRATCANS 1: GG1, PSA < 10 ng/mL, cT1c-cT2, and PSA density < 0.15 ng/mL^2^.STRATCANS 2:

GG1, PSA < 10 ng/mL, cT1-cT2, and PSA density ≥ 0.15 ng/mL^2^.GG1, PSA 10–20 ng/mL, cT1-cT2, and PSA density < 0.15 ng/mL^2^.GG2, PSA < 10 ng/mL, cT1-cT2, and PSA density < 0.15 ng/mL^2^.

3.STRATCANS 3:

GG1, PSA 10–20 ng/mL, cT1-cT2, and PSA density ≥ 0.15 ng/mL^2^.GG2, PSA < 10 ng/mL, cT1-cT2, and PSA density ≥ 0.15 ng/mL^2^.STRATCANS tiers can be calculated using the webtool at https://www.stratcans.com (accessed on 5 September 2025).

### 2.3. Study Objectives

The primary objective was to evaluate the association of STRATCANS with PC biopsy upgrading and time to definitive treatment. Primary outcomes included (1) biopsy upgrading to ≥GG3, (2) any biopsy upgrading (GG1 to ≥GG2, or GG2 to ≥GG3), and (3) time until definitive treatment. In the original STRATCANS studies, a rise in PSA without pathological change was not considered progression [22]. Similarly, in our study, progression was defined as pathologic upgrading and did not rely on PSA changes.

Our secondary objectives assessed whether having (1) an MRI before the diagnostic biopsy, (2) an MRI within 6 months after the diagnostic biopsy, or (3) genomic classifier (GC) testing modified the effect of STRATCANS on upgrade-free and treatment-free survival. Additional details regarding these analyses are provided in the Appendix A.

### 2.4. Statistical Analysis

Clinical, demographic, and oncologic characteristics were summarized as medians with interquartile ranges for continuous variables and counts with proportions for categorical variables. Baseline categorical measures and continuous measures were compared between STRATCANS tiers with the Chi-squared test and Kruskal–Wallis rank-sum test, respectively. For the primary objectives, Kaplan–Meier curves were used to graph time-to-event outcomes, with differences assessed using the log-rank test. The Benjamini–Yekutieli procedure was used to correct for multiple pairwise comparisons within each outcome. Patients in the upgrading analysis had at least one surveillance biopsy and were right-censored at the date of their last surveillance biopsy. All patients were included in the time to definitive treatment analysis. Patients remaining on AS were right-censored at the date of last clinical entry in the registry.

For secondary objectives, nine multivariable proportional hazards Cox models were fit, adjusting for patient demographics and clinical factors. Three models were fit for each primary outcome, incorporating interaction terms for STRATCANS and the prognostic tests of interest: pre-biopsy MRI, post-biopsy MRI, and GC testing. Hazard ratio effect sizes were plotted by outcome and prognostic test of interest, with detailed statistical analysis shown in the Appendix A [25,26].

## 3. Results

### 3.1. Demographics

Of 7578 men, 4009 met criteria for STRATCANS 1 (53%), 2732 for STRATCANS 2 (36%), and 837 for STRATCANS 3 (11%). The biopsy reclassification analysis included 2163, 1428, and 390 men with STRATCANS 1, 2, and 3, respectively. Median age of the cohort was 66 (IQR: 61–71), median PSA was 5.5 (IQR: 4.4–7.2), and most patients had GG1 disease (85%, Table 1 and Table 2). Median follow-up among patients without an event was 18 months (IQR: 12–31) for ≥GG3 upgrading, 18 months (IQR: 12–32) for any biopsy upgrading, and 29 months (IQR: 16–46) for time to definitive treatment.

### 3.2. Primary Objectives

Among 3981 patients in the reclassification cohort, there was a stepwise increase in the risk of disease progression to ≥GG3 across STRATCANS tiers, with an estimated 13% (95% CI: 11–15%), 33% (95% CI: 29–37%), and 53% (95% CI: 45–60%) of patients in STRATCANS 1, 2, and 3 progressing to ≥GG3 within 36 months after diagnosis, respectively (Figure 1a, *p* < 0.001). Since STRATCANS 2 and 3 included both GG1 and GG2 patients, we separated these into distinct cohorts based on GG (Figure 1b). We noted that STRATCANS 2/GG1 and STRATCANS 2/GG2 had similar risks of progression to ≥GG3, with approximately 32% (95% CI: 28–36%) and 36% (95% CI: 28–43%) of patients upgraded within 36 months after diagnosis (*p* > 0.9). Men with STRATCANS 3/GG1 and STRATCANS 3/GG2 also had a similar 36-month risk of progression to ≥GG3 (52% vs. 54%, *p* = 0.8).

The risk of any biopsy upgrading varied across STRATCANS tiers but did not increase in a stepwise manner. 44% (95% CI: 41–46%), 54% (95% CI: 51–58%), and 49% (95% CI: 41–56%) of patients in STRATCANS 1, 2, and 3, respectively, were upgraded to a higher GG on biopsy within 36 months after diagnosis (Figure 2a, *p* < 0.001). More men in STRATCANS 2/GG1 were upgraded at 36 months (61%, 95% CI: 56–64%) compared with STRATCANS 2/GG2 (34%, 95% CI: 27–41%, *p* < 0.001; Figure 2b). This was also the case among men in STRATCANS 3/GG1 (75%, 95% CI: 61–84%) vs. STRATCANS 3/GG2 (53%, 95% CI: 35–66%, *p* = 0.014).

STRATCANS was significantly associated with time to definitive treatment (Figure 3a, *p* < 0.001), with 16% (95% CI: 15–18%), 28% (95% CI: 26–30%), and 35% (95% CI: 31–39%) of men in STRATCANS 1, 2, and 3, respectively, transitioning from AS to definitive treatment within 36 months after diagnosis. Men in STRATCANS 2/GG1 and STRATCANS 2/GG2 had similar time to definitive treatment (28%, 95% CI: 25–30% vs. 29%, 95% CI: 25–33% at 3 years, *p* > 0.9) while men in STRATCANS 3/GG1 had less definitive treatment compared with STRATCANS 3/GG2 (30%, 95% CI: 24–35% vs. 41%, 95% CI: 35–46% at 3 years, *p* = 0.006; Figure 3b).

### 3.3. Secondary Objectives

Appendix A show forest plots demonstrating the effect estimates for each combination of STRATCANS tier and prognostic test of interest. Pre- and post-biopsy MRI did not significantly modify the effect of STRATCANS on ≥GG3 or any biopsy upgrading. However, both pre- and post-biopsy MRI significantly modified STRATCANS’ effect on time to definitive treatment (interaction terms: *p* < 0.001 and *p* = 0.003, respectively). GC testing did not significantly modify the effect of STRATCANS on ≥GG3 upgrading but did modify the effect for any biopsy upgrading (interaction term: *p* = 0.026). GC had no significant impact on the effect of STRATCANS with the time to definitive treatment outcome. A more detailed explanation of the results and interpretation of the forest plots is available in the Appendix A.

## 4. Discussion

STRATCANS was developed as a tool to risk-stratify men on AS for PC into tiered groups based on their likelihood of disease progression to provide a framework for risk-based follow-up protocol. While previous studies have looked at how STRATCANS can be used to guide personalized follow-up strategies for men on AS, its applicability to a large and diverse cohort of men on AS has not been well evaluated.

We found that STRATCANS was indeed associated with the risk of biopsy upgrading and treatment-free survival. As expected, men in higher STRATCANS tiers had an increased risk of upgrading to ≥GG3 disease. Interestingly, when breaking down patients in STRATCANS 2 and 3 into those with GG1 and GG2, patients with STRATCANS 2/GG1 and STRATCANS 2/GG2 had similar upgrading rates to ≥GG3, as did those with STRATCANS 3/GG1 and STRATCANS 3/GG2. This reflects the well-documented predictive value of PSA and PSA density on disease progression for men on AS [27,28,29,30]. This further demonstrates the strength of STRATCANS over initial GG alone, as STRATCANS 2 and 3, regardless of initial GG, were significantly associated with upgrading to ≥GG3, a meaningful endpoint which is a well-accepted trigger for ending AS and transitioning to definitive therapy.

In contrast, STRATCANS did not show an incremental trend for the outcome of any biopsy upgrading, with the 3-year risk remaining ~50% across all tiers. This is likely because any biopsy upgrading includes the upgrading of patients from GG1 to GG2, along with GG1 to ≥GG3 and GG2 to ≥GG3. Thus, this increases the number of patients in STRATCANS 2 and 3 who had GG1 disease and were upgraded to GG2, even though that may not have changed management. This becomes clear when we examine the breakdown of any biopsy upgrading, stratified by GG—the two groups with the highest risk of upgrading were patients in STRATCANS 2/GG1 (61%) and STRATCANS 3/GG1 (75%). This underscores the importance of using a more clinically relevant endpoint such as ≥GG3 or high-volume GG2 disease, where a change in management would be indicated.

While AS is appropriate for the vast majority of patients with GG1 and many with GG2 disease, accurately grouping AS patients by STRATCANS (or GG) depends on multiple factors, including the number and type of prior biopsies. For example, having multiple prior biopsies, pre-biopsy MRI, or fusion biopsies would enhance PC detection, likely reducing the frequency of upgrades to GG2 disease. This study looked at a heterogeneous set of patients, many of whom did not have a pre-biopsy MRI and may have been initially misclassified. Notably, Dr. Gnanapragasam’s group did report an association between STRATCANS and any objective progression, though this included change in MRI, any biopsy upgrading, or higher core volume. Furthermore, their study sample heavily utilized pre-biopsy MRI which likely minimized initial misclassification [17,18]. With this in mind, we continue to advocate for the consideration of AS in patients who initially present with GG2 disease, as well as those who progress to GG2 while on AS.

Time to definitive treatment was also associated with STRATCANS, with higher tiers being more likely to transition from AS to definitive treatment. Interestingly, when STRATCANS was stratified by initial GG, patients with STRATCANS 3/GG1 had higher treatment-free survival rates than those with STRATCANS 3/GG2, despite similar progression rates to ≥GG3. Unlike upgrading rates, treatment-free survival is a more subjective outcome, determined through shared decision-making between the patient and physician. The uncertainty of GG2 disease may have prompted the patient and/or clinician to opt for definitive therapy, despite having similar upgrading rates compared to GG1 patients with high PSA and/or PSA density.

STRATCANS was developed on a cohort where pre-biopsy MRI is heavily utilized, whereas in our cohort, only 13% of men had MRI before their diagnostic biopsy. An interesting finding was that MRI prior to diagnostic biopsy did not influence the effect of STRATCANS on ≥GG3 upgrading or any biopsy upgrading. Yet, MRI results significantly modified the association of STRATCANS with time to definitive treatment, suggesting MRI influences urologists’ and patients’ subjective decision-making in choosing between treatment or continued surveillance. While MRI remains the most widely used imaging modality in AS, its role is limited to anatomic and structural assessment only, and in the U.S., more advanced imaging approaches such as PSMA PET/CT scans or bone scintigraphy are generally reserved for higher-risk patients rather than those on surveillance. This underscores that STRATCANS was designed to function independently of such advanced imaging, ensuring broader applicability across healthcare systems with varying resources. In contrast, GC did significantly modify the association of STRATCANS with any biopsy upgrading. This supports emerging evidence that GC may provide marginal additional value in risk stratification beyond clinical factors for predicting future upgrading during AS [31].

Our study provides several novel contributions to the current STRATCANS literature. First, this analysis represents the largest cohort to date in which STRATCANS has been evaluated, including more than 7500 men [17,18]. Second, the MUSIC registry offers the most racially, socioeconomically, and geographically diverse cohort studied, encompassing patients from urban, suburban, and rural communities across academic and community-based practices. This diversity also extends to insurance coverage, with patients represented across private, public, and uninsured groups, further reflecting the real-world complexity of the U.S. healthcare system. In contrast, prior STRATCANS publications were predominantly limited to White European and Caucasian patients that did not emphasize diversity or even report race or ethnicity in their demographics. Third, the heterogeneity of diagnostic pathways in MUSIC, marked by variation in MRI use, biopsy technique, and absence of centralized pathology and radiology review, contrasts with the highly standardized settings of earlier studies, where practices in these European cohorts heavily utilized pre-biopsy MRI. Finally, this is the first study to evaluate STRATCANS in the context of GC and variable MRI use. We found that GC modified STRATCANS’ prognostic ability of any biopsy upgrading, while MRI significantly influenced treatment decision-making, even though neither altered the model’s prognostic ability for upgrading to ≥GG3 disease. Together, these findings demonstrate that STRATCANS not only reproduces results from prior studies, but also extends them by integrating traditional clinical and pathological data with imaging and molecular biomarkers, and by establishing its performance in a large and diverse real-world cohort. Our findings demonstrate that STRATCANS, when applied at the time of entry into AS, can effectively risk-stratify biopsy upgrading and treatment-free survival in a heterogeneous population. Moreover, this can be achieved without incurring any additional costs or resources, as each tier criterion relies on routine diagnostic data. This is exemplified by a prior European cohort study that prospectively modeled STRATCANS against current NICE guideline recommendations for AS, which include annual Digital Rectal Exam (DRE), repeat MRI at 12 months, and similar PSA intervals [18]. In their analysis, they found that STRATCANS reduced clinic visits by 22% and MRI scans by 42% in the first year, with estimated cost savings of £1518 per 100 men and £6027 per 100 men in outpatient visits and MRI costs, respectively. Because DRE is not part of the STRATCANS protocol, all follow-up appointments were conducted remotely, further reducing clinical burden. These findings support a risk-adapted approach for AS follow-up which could decrease related costs and clinical burden, while also helping harmonize follow-up schedules across diverse practices.

There are several limitations to our study. First, as a registry-based retrospective study, this inherently carries the risk of confounding and bias. Biopsies were performed by urologists of varying skill and training levels, with no standardized method utilized across physicians in MUSIC, and pathology and radiology lacked central review. This variability in biopsy methods, operator experience, and non-uniform pathology and imaging review may have affected both baseline risk classification and outcome ascertainment. Furthermore, only 13% of men in our cohort had an MRI-informed diagnostic biopsy, which may have also led to misclassification of patients at entry. Next, the primary outcomes we investigated (≥GG3 upgrading, any biopsy upgrading, and time until definitive treatment) were only followed out for 60 months with significant dropouts by 48 months, limiting long-term inference. Longer follow-up is still needed to establish the long-term outcomes and predictive validity of STRATCANS in men on AS. Finally, the outcome of time to definitive treatment was determined by shared decision-making between the patient and physician, rather than a standardized set of criteria. This may have introduced variability in the timing of treatment initiation across practices.

Despite these limitations, STRATCANS maintained prognostic discrimination under these real-world conditions, highlighting its robustness and external validity across diverse U.S. cohorts and practice settings. Overall, our findings suggest that STRATCANS is both reliable and broadly generalizable, supporting its potential role in guiding risk-adapted follow-up for men on AS.

## 5. Conclusions

STRATCANS was associated with the risk of progression to ≥GG3 disease and time to definitive treatment in men on AS within MUSIC in a stepwise manner. Additionally, STRATCANS appears superior to GG alone, as patients with different GGs within the same STRATCANS tier had similar risks of upgrading to ≥GG3 disease. These findings align with those in previous studies and favor moving towards a risk-adapted follow-up strategy for monitoring men on AS.

## Figures and Tables

**Figure 1 cancers-17-03032-f001:**
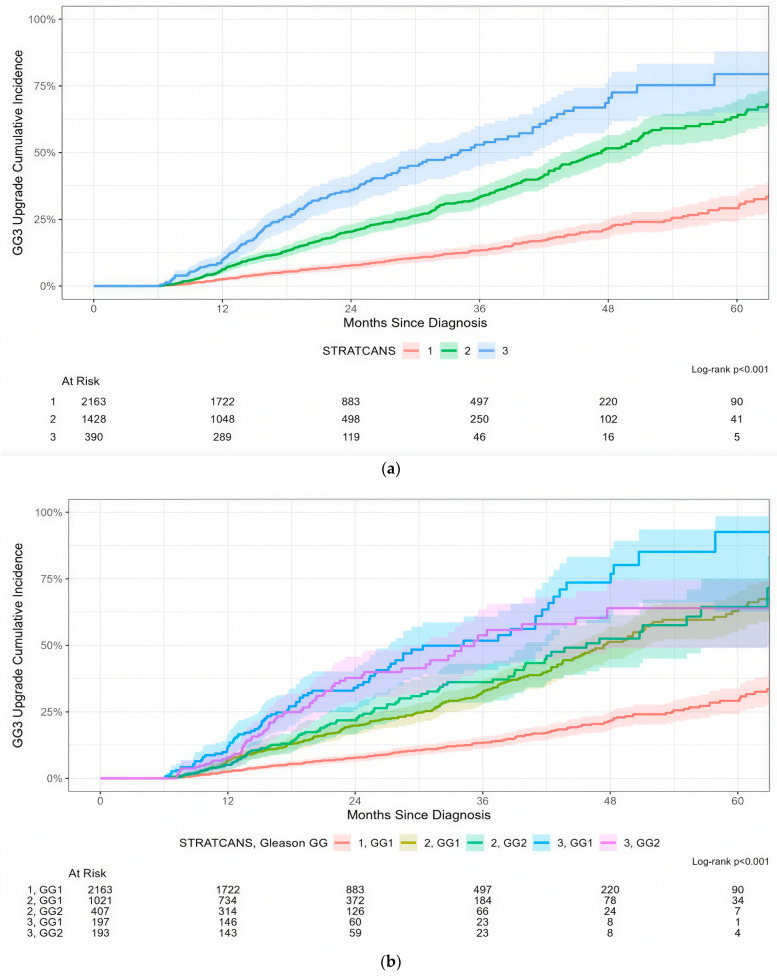
(**a**) Time to biopsy upgrading to ≥GG3 prostate cancer according to STRATCANS group. (**b**) Time to biopsy upgrading to ≥GG3 prostate cancer according to STRATCANS group, with further stratification by grade group at initial biopsy (GG1 or GG2).

**Figure 2 cancers-17-03032-f002:**
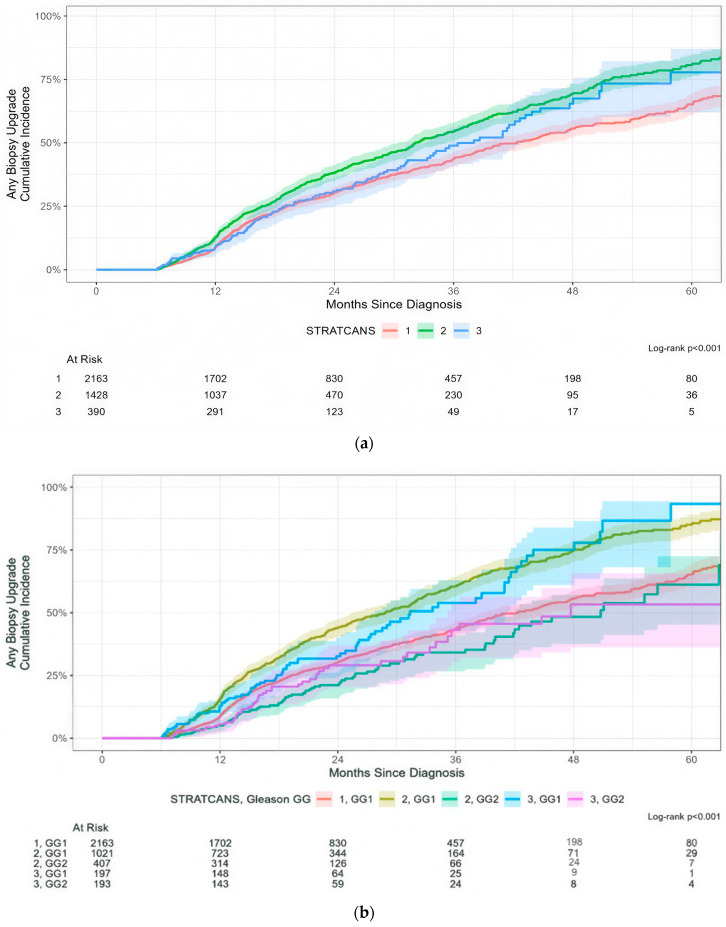
(**a**) Time to any biopsy upgrading in each STRATCANS group. (**b**) Time to any biopsy upgrading in each STRATCANS group, with further stratification by grade group (GG1 or GG2).

**Figure 3 cancers-17-03032-f003:**
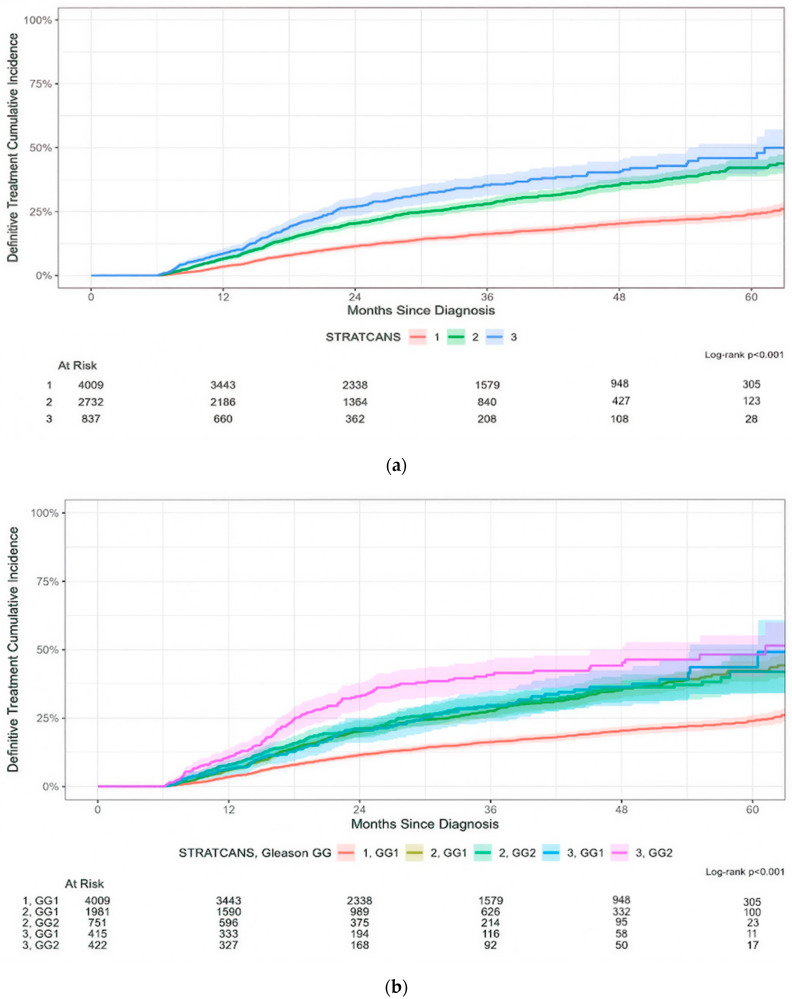
(**a**) Time to definitive treatment in each STRATCANS group. (**b**) Time to definitive treatment in each STRATCANS group, which is further stratified by grade group (GG1 or GG2).

**Table 1 cancers-17-03032-t001:** Patient demographics, clinical factors, and oncologic parameters of patients included in the time to definitive treatment analysis. Continuous variables are shown as medians with interquartile ranges (IQRs). Categorical measures are compared with the Chi-squared test and continuous variables with the Kruskal–Wallis rank-sum test.

	STRATCANS 1 *n* = 4009	STRATCANS 2 *n* = 2732	STRATCANS 3 *n* = 837	*p*-Value
Age	66 (61, 70)	66 (61, 71)	67 (62, 72)	<0.001
Race				<0.001
White	3227 (80%)	2156 (79%)	629 (75%)	
African American	367 (9.2%)	316 (12%)	121 (14%)	
Other	91 (2.3%)	67 (2.5%)	26 (3.1%)	
Unknown	324 (8.1%)	193 (7.1%)	61 (7.3%)	
Charlson Score				0.026
0	2980 (77%)	1947 (74%)	581 (72%)	
1	588 (15%)	427 (16%)	152 (19%)	
≥2	316 (8.1%)	243 (9.3%)	73 (9.1%)	
Missing	125	115	31	
Family History of Prostate Cancer	1210 (31%)	732 (28%)	218 (27%)	0.003
Missing	144	98	36	
Insurance Type				<0.001
None	33 (0.8%)	18 (0.7%)	3 (0.4%)	
Private	2242 (56%)	1407 (52%)	402 (48%)	
Public	1709 (43%)	1284 (47%)	425 (51%)	
Missing	25	23	7	
PSA at Dx	4.84 (4.00, 5.94)	6.10 (4.89, 7.70)	9.82 (6.51, 12.20)	<0.001
PSA Density at Dx	0.09 (0.07, 0.12)	0.17 (0.13, 0.22)	0.22 (0.18, 0.30)	<0.001
Gleason Grade Group at Dx				<0.001
3 + 3	4009 (100%)	1981 (73%)	415 (50%)	
3 + 4	0 (0%)	751 (27%)	422 (50%)	
Clinical T-Stage at Dx				0.06
T1	3618 (90%)	2484 (91%)	777 (93%)	
T2	391 (9.8%)	248 (9.1%)	60 (7.2%)	

IQRs, interquartile ranges; PSA, prostate specific antigen; Dx, diagnosis.

**Table 2 cancers-17-03032-t002:** Patient demographics, clinical factors, and oncologic parameters of patients included in the biopsy upgrading cohort. Continuous variables are shown as medians with IQRs. Categorical measures are compared with the Chi-squared test and continuous variables with the Kruskal–Wallis rank-sum test.

	STRATCANS 1 *n* = 2163	STRATCANS 2 *n* = 1428	STRATCANS 3 *n* = 390	*p*-Value
Age	65 (61, 70)	66 (60, 70)	66 (61, 71)	0.039
Race				0.017
White	1734 (80%)	1107 (78%)	307 (79%)	
African American	199 (9.2%)	176 (12%)	45 (12%)	
Other	48 (2.2%)	42 (2.9%)	14 (3.6%)	
Unknown	182 (8.4%)	103 (7.2%)	24 (6.2%)	
Charlson Score				0.005
0	1652 (78%)	1016 (74%)	273 (72%)	
1	285 (14%)	231 (17%)	72 (19%)	
≥2	173 (8.2%)	132 (9.6%)	34 (9.0%)	
Missing	53	49	11	
Family History of Prostate Cancer	695 (33%)	373 (27%)	111 (29%)	<0.001
Missing	78	50	13	
Insurance Type				
None	12 (0.6%)	11 (0.8%)	1 (0.3%)	
Private	1290 (60%)	791 (56%)	202 (52%)	
Public	849 (39%)	616 (43%)	184 (48%)	
Missing	12	10	3	
PSA at Dx	4.79 (3.93, 5.74)	5.97 (4.76, 7.50)	10.05 (6.55, 12.08)	<0.001
PSA Density at Dx	0.09 (0.07, 0.12)	0.17 (0.13, 0.22)	0.23 (0.18, 0.31)	<0.001
Gleason Grade Group at Dx				<0.001
3 + 3	2163 (100%)	1021 (71%)	197 (51%)	
3 + 4	0 (0%)	407 (29%)	193 (49%)	
Clinical T-Stage at Dx				0.094
T1	1970 (91%)	1302 (91%)	368 (94%)	
T2	193 (8.9%)	126 (8.8%)	22 (5.6%)	

IQRs, interquartile ranges; PSA, prostate specific antigen; Dx, diagnosis.

## Data Availability

Raw data are not publicly available due to MUSIC data sharing agreements. Statistical code can be provided upon reasonable request.

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
