# Peer review of "Application of the STRATCANS Criteria to the MUSIC Prostate Cancer Active Surveillance Cohort: A Step Towards Risk-Stratified Active Surveillance"

_cancers, 2025, doi:10.3390/cancers17183032_

Round 1
Reviewer 1 Report
Comments and Suggestions for Authors
I have only minor comments to upgrade the manuscript:
- In the Title, at least the word "prostate cancer" should be mentioned, and AS should be spelled out.
- In the Simple Summary, STRATCANS and GG should be spelled out on their first appearance, followed by the abbreviated form in parentheses, to provide clarity for the reader.
- In the Abstract, GG should be spelled out.
Regarding the Secondary Objectives (Materials and Methods):
The authors use pre- and post-biopsy MRI in this cohort. As "predicted", pre- and post-biopsy MRI did not significantly modify the effect of STRATCANS on ≥ GG3 or any biopsy upgrading. In my opinion, this is because MRI provides only anatomical/structural images. In the US, Ga-68 PSMA PET/CT imaging is a common and widely available diagnostic tool. Even bone scintigraphy is often regarded as helpful in monitoring or in AS. We understand that STRATCANS may be designed for global use (and not every country can provide PSMA PET/CT), but bone scintigraphy is inexpensive and widely available globally. I think it should be mentioned, at the very least, in the design considerations of STRATCANS.
The secondary objective findings (including the images) may be appropriate to mention in the main text, as imaging (in this study, MRI) is an integral method used in AS.
Author Response
Comments and Suggestions for Authors
I have only minor comments to upgrade the manuscript:
Comment 1: In the Title, at least the word "prostate cancer" should be mentioned, and AS should be spelled out.
RESPONSE 1: Thank you for your feedback. We have accordingly updated our title to read: “Application of the STRATCANS Criteria to the MUSIC Prostate Cancer Active Surveillance Cohort: A Step Towards Risk-Stratified Active Surveillance.”
Comment 2: In the Simple Summary, STRATCANS and GG should be spelled out on their first appearance, followed by the abbreviated form in parentheses, to provide clarity for the reader.
RESPONSE 2: Thank you for your feedback. We have accordingly updated Simple Summary with these changes.
Comment 3: In the Abstract, GG should be spelled out.
RESPONSE 3: Thank you for your feedback. We have updated this in our Abstract.
Comment 4:
Regarding the Secondary Objectives (Materials and Methods):
The authors use pre- and post-biopsy MRI in this cohort. As "predicted", pre- and post-biopsy MRI did not significantly modify the effect of STRATCANS on ≥ GG3 or any biopsy upgrading. In my opinion, this is because MRI provides only anatomical/structural images. In the US, Ga-68 PSMA PET/CT imaging is a common and widely available diagnostic tool. Even bone scintigraphy is often regarded as helpful in monitoring or in AS. We understand that STRATCANS may be designed for global use (and not every country can provide PSMA PET/CT), but bone scintigraphy is inexpensive and widely available globally. I think it should be mentioned, at the very least, in the design considerations of STRATCANS.
The secondary objective findings (including the images) may be appropriate to mention in the main text, as imaging (in this study, MRI) is an integral method used in AS.
RESPONSE 4: We thank the reviewer for this thoughtful comment. We agree that MRI has limitations, and ongoing research is elucidating the role of PSMA PET/CT to complement MRI in the diagnosis and monitoring of men with prostate cancer. Currently, in the United States, PSMA PET/CT is only indicated for men with unfavorable intermediate-risk, high-risk, or very high–risk prostate cancer, and not for low-risk or favorable intermediate–risk men who comprise the active surveillance cohort studied here. Additionally, the role of PSMA PET/CT in active surveillance is still being defined and is not yet ready for broad clinical application in this setting. Similarly, bone scintigraphy is also indicated for metastatic evaluation in higher-risk patients and is not recommended for men with low-risk or favorable intermediate-risk disease on active surveillance. We have expanded the Discussion to include the role and limitations of MRI, with mention of PSMA PET/CT and bone scintigraphy, while keeping the detailed analyses and figures in the supplemental materials to maintain clarity and focus in the main text.
Page 10, Discussion paragraph 6, Lines 274-279: While MRI remains the most widely used imaging modality in AS, its role is limited to anatomic and structural assessment only, and in the U.S., more advanced imaging approaches such as PSMA PET/CT or bone scintigraphy are generally reserved for higher-risk patients rather than those on surveillance. This underscores that STRATCANS was designed to function independently of such advanced imaging, ensuring broader applicability across health care systems with varying resources.

Reviewer 2 Report
Comments and Suggestions for Authors
With great interest, I reviewed the manuscript evaluating the applicability of the STRATCANS risk-stratification tool within the MUSIC Active Surveillance (AS) cohort. The study addresses an important and clinically relevant question: how to personalize surveillance intensity in prostate cancer patients on AS, balancing safety with reduced patient and system burden. The manuscript is well written and based on a large, high-quality registry dataset. However, several issues limit the strength, novelty, and generalizability of the conclusions.
Major comments:
Novelty and incremental value: STRATCANS has already been validated in UK and Spanish cohorts. The current analysis largely confirms previously published findings. The authors should better articulate what is novel here: is it the diversity of the MUSIC cohort, practice heterogeneity, or integration with genomic classifiers and MRI? Without a clearer positioning, the study risks appearing confirmatory rather than innovative.
Retrospective design and confounding: The registry-based, retrospective design carries inherent risks of confounding and bias. For example, only 13% of patients had pre-biopsy MRI, pathology was not centrally reviewed, and biopsy methods varied substantially. These limitations should be discussed more prominently, especially as they might affect both baseline risk classification and outcome ascertainment.
Definition of endpoints: The use of “any upgrading” as a co-primary outcome dilutes the clinical significance of results, since many upgrades from GG1→GG2 do not alter management. The authors should emphasize ≥GG3 upgrading or clinically meaningful endpoints (e.g., high-volume GG2) as more appropriate triggers for intervention.
Follow-up duration: Median follow-up of 18–29 months is relatively short to assess outcomes such as upgrading and treatment transition in AS. The authors should explicitly acknowledge that longer follow-up is needed to establish long-term predictive validity of STRATCANS in this setting.
Integration with modern diagnostics: One of the most important aspects is how STRATCANS interacts with MRI and genomic classifiers (GC). While the supplementary analyses are interesting , they are underdeveloped in the main text. Stronger emphasis should be placed on how STRATCANS might be adapted in an era where pre-biopsy MRI and molecular tools are increasingly standard of care.
Clinical applicability: The manuscript would benefit from more discussion on the practical implementation of STRATCANS: How would it reduce biopsy burden in a real-world US setting? Could it harmonize follow-up schedules across practices? What are the cost-effectiveness implications?
Minor comments:
Abstract: The abstract is clear but somewhat long; consider tightening while maintaining a structured format.
Figures and tables: Kaplan-Meier plots are appropriate but should include numbers at risk for greater clarity.
Language: Generally good, though some sentences are dense; minor stylistic polishing would improve readability.
References: Ensure inclusion of the most recent validation papers and AS guideline updates (e.g., EAU 2024, AUA/ASTRO 2022).
Supplementary material: Some results (e.g., effect modification by MRI/GC) are essential and might be summarized in the main manuscript rather than relegated to the supplement.
Author Response
Comments and Suggestions for Authors
With great interest, I reviewed the manuscript evaluating the applicability of the STRATCANS risk-stratification tool within the MUSIC Active Surveillance (AS) cohort. The study addresses an important and clinically relevant question: how to personalize surveillance intensity in prostate cancer patients on AS, balancing safety with reduced patient and system burden. The manuscript is well written and based on a large, high-quality registry dataset. However, several issues limit the strength, novelty, and generalizability of the conclusions.
Major comments:
Comment 1: Novelty and incremental value: STRATCANS has already been validated in UK and Spanish cohorts. The current analysis largely confirms previously published findings. The authors should better articulate what is novel here: is it the diversity of the MUSIC cohort, practice heterogeneity, or integration with genomic classifiers and MRI? Without a clearer positioning, the study risks appearing confirmatory rather than innovative.
REPONSE 1: We thank the reviewer for this important comment. We agree that the novelty of our study needed clearer articulation, and we have revised the Discussion section accordingly. In the updated text, we emphasize that this is the largest cohort to utilize and validate the STRATCANS criteria to date, including over 7,500 men. We also highlight that our cohort demonstrates substantial racial and socioeconomic diversity across a wide range of practice settings, in contrast to prior STRATCANS publications which were conducted in relatively uniform practice environments with less diverse populations. Finally, we note that our study is the first to evaluate STRATCANS alongside genomic classifiers and heterogeneous MRI use. Together, these points position our study as more than confirmatory, demonstrating that STRATCANS is not only validated but also broadly generalizable and integrated with modern imaging and molecular tools in diverse clinical practice.
Page 10, Discussion paragraph 7, Lines 284-304: Our study provides several novel contributions to the current STRATCANS literature. First, this analysis represents the largest cohort to date in which STRATCANS has been evaluated, including more than 7,500 men [18, 19]. Second, the MUSIC registry offers the most racially, socioeconomically, and geographically diverse cohort studied, encompassing patients from urban, suburban, and rural communities across academic and community-based practices. This diversity also extends to insurance coverage, with patients represented across private, public, and uninsured groups, further reflecting the real-world complexity of the U.S. healthcare system. In contrast, prior STRATCANS publications were predominantly limited to White European and Caucasian patients that did not emphasize diversity or even report race or ethnicity in their demographics. Third, the heterogeneity of diagnostic pathways in MUSIC, marked by variation in MRI use, biopsy technique, and absence of centralized pathology or radiology review, contrasts with the highly standardized settings of earlier studies, where practices in these European cohorts heavily utilized pre-biopsy MRI. Finally, this is the first study to evaluate STRATCANS in the context of GC and variable MRI use. We found that GC modified STRATCANS’ prognostic ability of any biopsy upgrading, while MRI significantly influenced treatment decision-making, even though neither altered the model’s prognostic ability for upgrading to ≥ GG3 disease. Together, these findings demonstrate that STRATCANS not only reproduces results from prior studies but also extends them by integrating traditional clinical and pathological data with imaging and molecular biomarkers, and by establishing its performance in a large and diverse real-world cohort.
Comment 2: Retrospective design and confounding: The registry-based, retrospective design carries inherent risks of confounding and bias. For example, only 13% of patients had pre-biopsy MRI, pathology was not centrally reviewed, and biopsy methods varied substantially. These limitations should be discussed more prominently, especially as they might affect both baseline risk classification and outcome ascertainment.
RESPONSE 2: We thank the reviewer for their suggestion. In response, we have revised the limitations section to more prominently acknowledge the limitations inherent to a retrospective, registry-based design. Specifically, we now note that variability in biopsy methods, operator experience, and absence of centralized pathology and radiology review may have influenced both baseline risk classification and outcome ascertainment. We also highlight that the small proportion of men (13%) who underwent an MRI-informed diagnostic biopsy could also have contributed to misclassification. While these factors may reduce internal validity and introduce potential sources of confounding and bias, they also underscore the real-world generalizability of our findings: STRATCANS maintained prognostic discrimination despite this heterogeneity, emphasizing its validity across diverse cohorts and practice settings.
Page 11, Discussion paragraph 8, lines 319-326: First, as a registry-based retrospective study, this inherently carries the risk of confounding and bias. Biopsies were performed by urologists of varying skill and training levels, with no standardized method utilized across physicians in MUSIC, and pathology and radiology lacked central review. This variability in biopsy methods, operator experience, and non-uniform pathology or imaging review may have affected both baseline risk classification and outcome ascertainment. Furthermore, only 13% of men in our cohort had an MRI-informed diagnostic biopsy, which may have also led to misclassification of patients at entry.
Page 11, Discussion paragraph 9, Lines 334-338: Despite these limitations, STRATCANS maintained prognostic discrimination under these real-world conditions, highlighting its robustness and external validity across diverse U.S. cohorts and practice settings. Overall, our findings suggest that STRATCANS is both reliable and broadly generalizable, supporting its potential role in guiding risk-adapted follow-up for men on AS.
Comment 3: Definition of endpoints: The use of “any upgrading” as a co-primary outcome dilutes the clinical significance of results, since many upgrades from GG1→GG2 do not alter management. The authors should emphasize ≥GG3 upgrading or clinically meaningful endpoints (e.g., high-volume GG2) as more appropriate triggers for intervention.
RESPONSE 3: We thank the reviewer for this comment. We believe that this is already addressed in the third paragraph of the Discussion, where we explain why “any upgrading” did not show an incremental trend across STRATCANS tiers, and also highlight how upgrading to ≥GG3 is a more clinically relevant endpoint, as it is more likely to indicate a change in management.
Comment 4: Follow-up duration: Median follow-up of 18–29 months is relatively short to assess outcomes such as upgrading and treatment transition in AS. The authors should explicitly acknowledge that longer follow-up is needed to establish long-term predictive validity of STRATCANS in this setting.
RESPONSE 4: We thank the reviewer for this comment. While we had noted in the limitations that our 60-month follow-up with significant dropouts by 48 months limited long-term inference, we agree that a clearer acknowledgment of the need for longer follow-up should have been made. Accordingly, we have now added a statement noting this.
Page 11, Discussion paragraph 8, Lines 329–330: Next, the primary outcomes we investigated (≥ GG3 upgrading, any biopsy upgrading, and time until definitive treatment) were only followed out for 60 months with significant dropouts by 48 months, limiting long-term inference. Longer follow-up is still needed to establish the long-term outcomes and predictive validity of STRATCANS in men on AS
Comment 5: Integration with modern diagnostics: One of the most important aspects is how STRATCANS interacts with MRI and genomic classifiers (GC). While the supplementary analyses are interesting , they are underdeveloped in the main text. Stronger emphasis should be placed on how STRATCANS might be adapted in an era where pre-biopsy MRI and molecular tools are increasingly standard of care.
RESPONSE 5: We agree that the use of MRI and genomic classifiers is becoming increasingly relevant, and that their appropriate integration into clinical practice is the subject of ongoing research across multiple groups. As noted in our analyses, STRATCANS retained prognostic discrimination regardless of MRI or GC use, and the additional value of these tools was minimal. MRI primarily modified the association of STRATCANS with time to definitive treatment, suggesting its impact on subjective shared decision-making, while GC only modified the outcome of any biopsy upgrading. Given the already broad scope and length of the manuscript, we believe these data are best presented within the supplementary material and figures to maintain focus within the main manuscript text.
Comment 6: Clinical applicability: The manuscript would benefit from more discussion on the practical implementation of STRATCANS: How would it reduce biopsy burden in a real-world US setting? Could it harmonize follow-up schedules across practices? What are the cost-effectiveness implications?
RESPONSE 6: We thank the reviewer for raising these important questions. While we agree that analyses of healthcare utilization and cost-effectiveness are critical aspects of STRATCANS’ broader value, these remain important avenues for future research, as they are outside the scope of the current study. Prior work in European cohorts has demonstrated reduced financial and logistical burden when comparing STRATCANS with existing guideline-based follow-up, and we have now updated the manuscript to reference this in our Discussion section.
Page 11, Discussion paragraph 7, Lines 308-318: This is exemplified by a prior European cohort study that prospectively modelled STRATCANS against current NICE guideline recommendations for AS, which include annual Digital Rectal Exam (DRE), repeat MRI at 12 months, and similar PSA intervals [18]. In their analysis, they found that STRATCANS reduced clinic visits by 22% and MRI scans by 42% in the first year, with estimated cost savings of £1,518 per 100 men and £6,027 per 100 men in outpatient visits and MRI costs, respectively. Because DRE is not part of the STRATCANS protocol, all follow-up appointments were conducted remotely, further reducing clinical burden. These findings support a risk-adapted approach for AS follow-up which could decrease related costs and clinical burden, while also helping harmonize follow-up schedules across diverse practices.
Minor comments:
Comment 7: Abstract: The abstract is clear but somewhat long; consider tightening while maintaining a structured format.
RESPONSE 7: We thank Reviewer 2 for their suggestion regarding our abstract. However, Reviewer 3 thought our abstract was a concise and informative summary of the manuscript, and Reviewer 1 did not recommend any changes. While we are satisfied with the current abstract, we defer to the Editors’ expertise, should they wish us to further condense it to be more in line with Reviewer 2’s suggestion.
Comment 8: Figures and tables: Kaplan-Meier plots are appropriate but should include numbers at risk for greater clarity.
RESPONSE 8: We thank the reviewer for this comment. We note that our current Kaplan–Meier plots already include risk tables with the numbers at risk displayed directly below each graph.
Comment 9: Language: Generally good, though some sentences are dense; minor stylistic polishing would improve readability.
RESPONSE 9: We thank the reviewer for this comment. We have carefully reviewed the manuscript and believe the current language is clear and appropriate for the journal. We also note that no other reviewers raised concerns about language or sentence structure.
Comment 10: References: Ensure inclusion of the most recent validation papers and AS guideline updates (e.g., EAU 2024, AUA/ASTRO 2022).
RESPONSE 10: Thank you for your feedback. We have updated the manuscript to include several additional references, including the most recent EAU Prostate Cancer 2025 guidelines.
Comment 11: Supplementary material: Some results (e.g., effect modification by MRI/GC) are essential and might be summarized in the main manuscript rather than relegated to the supplement.
RESPONSE 11: We thank the reviewer for this comment. As discussed above, there is already a large amount of data included in the main text, and we prioritized presenting what we believe are the most important and novel aspects of STRATCANS in a diverse, real-world cohort. We feel that keeping the additional analyses on MRI and GC in the supplementary material maintains clarity and balance in the manuscript, while still ensuring that interested readers have full access to these results.

Reviewer 3 Report
Comments and Suggestions for Authors
highly interesting manuscript on one of the most controversial topics in PCa - optimal patient`s allocation for AS and optimal protocol, using risk-stratification criteria
Title - precisely depicting the essence of the study - No remarks
Abstract - concise review of the manuscript - No remarks
Introduction - in-depth presentation of the contemporary literature, and defining the gaps, that this research aims to fill - No remarks
Material and methods - detailed description of the sophisticated study protocol - No remarks
Results - row 149-150 - needs additional clarification of difference with time to definitive treatment cohort - second biopsy? secondary analysis of first biopsy? timing? - Major
discussion and conclusions - the authors elegantly implement their results into the contemporary literature, firmly substantiating their conclusions.
The limitations, that authors are citing, are inevitable for this type of validation in a real world cohort. Heterogeneity in this cohort could be interpreted even as advantage, as it is assuring for STRATCANS generalizability as a prognostic factor in AS for PCa. - No remarks
Author Response
Comments and Suggestions for Authors
Comment 1: highly interesting manuscript on one of the most controversial topics in PCa - optimal patient`s allocation for AS and optimal protocol, using risk-stratification criteria
RESPONSE 1: Thank you for your favorable impression of our study.
Comment 2: Title - precisely depicting the essence of the study - No remarks
RESPONSE 2: Thank you for your favorable review.
Comment 3: Abstract - concise review of the manuscript - No remarks
RESPONSE 3: Thank you for your favorable review.
Comment 4: Introduction - in-depth presentation of the contemporary literature, and defining the gaps, that this research aims to fill - No remarks
RESPONSE 4: Thank you for your favorable review.
Comment 5: Material and methods - detailed description of the sophisticated study protocol - No remarks
RESPONSE 5: Thank you for your favorable review.
Comment 6: Results - row 149-150 - needs additional clarification of difference with time to definitive treatment cohort - second biopsy? secondary analysis of first biopsy? timing? - Major
RESPONSE 6: We thank the reviewer for this feedback. As described in the Methods, the time to definitive treatment analysis included all men on active surveillance, whereas the biopsy upgrading/reclassification analyses were restricted to those with at least one surveillance biopsy. As noted in the fifth paragraph of the Discussion, time to definitive treatment was determined through shared decision-making, making it a subjective outcome. However, for further clarification, we have now added this point to the limitations section as well.
Page 11, Discussion paragraph 8, Lines 330-333: Finally, the outcome of time to definitive treatment was determined by shared decision-making between the patient and physician, rather than a standardized set of criteria. This may have introduced variability in the timing of treatment initiation across practices.
Comment 7: discussion and conclusions - the authors elegantly implement their results into the contemporary literature, firmly substantiating their conclusions.
RESPONSE 7: Thank you for your favorable review.
Comment 8: The limitations, that authors are citing, are inevitable for this type of validation in a real world cohort. Heterogeneity in this cohort could be interpreted even as advantage, as it is assuring for STRATCANS generalizability as a prognostic factor in AS for PCa. - No remarks
RESPONSE 8: Thank you for your favorable review.

Round 2
Reviewer 3 Report
Comments and Suggestions for Authors
the authors has sufficiently taken into consideration this reviewer`s comments